# Competitive Sperm-Marked Beetles for Monitoring Approaches in Genetic Biocontrol and Studies in Reproductive Biology

**DOI:** 10.3390/ijms232012594

**Published:** 2022-10-20

**Authors:** Musa Dan’azumi Isah, Bibi Atika, Stefan Dippel, Hassan M. M. Ahmed, Ernst A. Wimmer

**Affiliations:** 1Department of Developmental Biology, Johann-Friedrich-Blumenbach-Institute for Zoology and Anthropology, Ernst-Caspari-Haus, GZMB, Georg-August-University Goettingen, Justus-von-Liebig-Weg 11, 37077 Goettingen, Germany; 2Department of Crop Science, Faculty of Agriculture, Wildlife and Forestry Resources Management, University of Calabar, Calabar P.M. B. 1152, Cross River State, Nigeria

**Keywords:** insect biotechnology, molecular entomology, pest management, sterile insect technique, sperm storage, transgenesis, *Tribolium castaneum*

## Abstract

Sperm marking provides a key tool for reproductive biology studies, but it also represents a valuable monitoring tool for genetic pest control strategies such as the sterile insect technique. Sperm-marked lines can be generated by introducing transgenes that mediate the expression of fluorescent proteins during spermatogenesis. The homozygous lines established by transgenesis approaches are going through a genetic bottleneck that can lead to reduced fitness. Transgenic SIT approaches have mostly focused on Dipteran and Lepidopteran pests so far. With this study, we provide sperm-marked lines for the Coleopteran pest model organism, the red flour beetle *Tribolium castaneum*, based on the *β2-tubulin* promoter/enhancer driving red (DsRed) or green (EGFP) fluorescence. The obtained lines are reasonably competitive and were thus used for our studies on reproductive biology, confirming the phenomenon of ‘last-male sperm precedence’ and that the spermathecae are deployed for long-term sperm storage, enabling the use of sperm from first mating events even after secondary mating events for a long period of time. The homozygosity and competitiveness of the lines will enable future studies to analyze the controlled process of sperm movement into the long-term storage organ as part of a post-mating cryptic female choice mechanism of this extremely promiscuous species.

## 1. Introduction

Studies on reproductive biology related to the mating behavior of polyandrous species, such as sperm storage, sperm precedence, sperm use, and sperm competition, can be improved by fluorescently marked sperm using transgenesis approaches [1] Such transgenic marking systems have been described for the mosquitoes *Anopheles stephensi* [2] and *Aedes aegypti* [3], as well as the Tephritid fruit flies *Ceratitis capitata* [4] and *Anastrepha suspensa* [5], and the cherry vinegar fly *Drosophila suzukii* [6]. All these systems are based on the endogenous spermatogenesis-specific *β2-tubulin* (*β2t*) promoter driving testis-specific expression of fluorescent markers, which enables an inheritable sex-specific marking. Since such systems allow for the detection of the sperm even in mated females [4], they could extend the abilities to also monitor the receptiveness of wild females in genetic insect control programs such as the sterile insect technique (SIT).

The SIT is a birth management practice geared toward the efficient control of insect pest populations, which represents a safe, species-specific, and environmentally friendly management strategy [7]. SIT has been used successfully to eradicate insect pests such as the Tsetse fly in Zanzibar, the New World Screwworm in North and Middle America, as well as in North Africa [8], and is used to manage several Tephritid fruit fly pests around the world [9]. SIT represents a classical genetic pest control method, which reduces a pest population by mass rearing and releasing reproductively sterile males that compete with wild males for wild females [10]. Besides the mass rearing, sterilization, and release of the pest species, monitoring is of major importance for this cost-intensive method. To calculate the ratio of released to wild insects, the data from field traps are used [11]. At present, mass-reared pupae are dusted with fluorescent dyes that are expensive, dangerous for human health, and error-prone [12]. Fluorescently marked sperm using transgenesis approaches could serve as an excellent alternative, which would extend the monitoring to the receptiveness of wild females regarding the released sterile males and thus their effectiveness in mating and sperm transfer [4].

There is a historical bias toward using SIT mostly in Diptera and Lepidoptera, which is probably based on both the large number of pest species in these orders and the biological basis for the implementation of SIT [13]. Despite the fact that the order Coleoptera also includes a high number of pest species, they have only been addressed to a lesser extent, probably because SIT seems impractical for those species that are highly destructive at the adult stage [10]. However, there are also quite a number of severe coleopteran pests, especially weevils, that do not cause much damage as adults, and for which SIT can, therefore, be considered [8]. Despite the fact that the red flour beetle *Tribolium castaneum* (Coleoptera: Tenebrionidae) does not serve as a target organism for SIT, we present here a proof of principle for transgenic sperm marking in this coleopteran model organism for development and pest biology [14]. We show that such transgenic lines are reasonably competitive and can be used for monitoring as well as for reproductive biology studies in this extremely promiscuous species showing cryptic female choice [15].

## 2. Results

### 2.1. Generation of Different Tribolium Castaneum Sperm-Marking Lines

To establish a sperm-marking system that drives fluorescent proteins specifically in the testis of *T. castaneum*, we identified the *β2t* gene (TC009035, GENBANK accession number XP_969993) in its genome [16]. The *β2t* gene codes for a 452 amino acid protein encoded by a 1347 bp Open Reading Frame that is bordered by a 121 bp 5′UTR and a 258 bp 3′UTR and contained on two exons that are interrupted by a single 46 bp intron (Figure 1A). To determine the regulatory elements that are needed for effective and efficient gene expression specifically in the *T. castaneum* testes, we tried to identify the conserved testis-specific regulatory elements: *β2t Upstream Element 1* (*β2UE1*), *β2UE2*, the 7bp initiator sequence, and the downstream element *β2tDE1*, which have been described as important for TATA-less promoters in *D. melanogaster* [17] and *D. suzuki* (6). However, we could not identify these elements in the *T. castaneum* genome. Therefore, we used a 1 Kb promoter/enhancer (P/E) region upstream of the ATG translation start codon of the spermatogenesis-specific gene *β2t* to drive the red fluorescing protein DsRed (*β2t*-DsRed-SV40; Figure 1B) or the green fluorescent protein EGFP (*β2t*-EGFP-SV40; Figure 1C). The constructs were then inserted into *piggyBac* transformation vectors [18,19] carrying different fluorescent markers driven by an eye-specific promoter 3xP3 [20].

To generate the transgenic lines expressing the fluorescent proteins in their testes, we performed *piggyBac*-based germline transformation [20]. For *β2t*-EGFP-SV40, we obtained three injected G_0_ giving rise to transgenic progeny (transformation rate 5.5%; Appendix A). Four F_1_ progeny lines were set up (Appendix A), two of which could be homozygoused (F37_F1 and F37_M2), and were kept and used for subsequent experiments. For *β2t*-DsRed-SV40, we obtained four injected G_0_ giving rise to transgenic progeny (transformation rate 4,9%; Appendix A). From them, 18 F_1_ progeny lines were set up (Appendix A), 2 of which were independent lines that could be homozygoused (M1_F1 and F1_M1), and were kept and used for subsequent experiments. The transgenic male and female individuals could be identified by their eye fluorescence, while the testis fluorescence was only detected in males restricted to the testes (Figure 1). To identify the genomic integration sites of these lines, inverse PCR was performed, which confirmed the independent integration of M1_F1 and F1_M1 (Appendix A).

To examine whether the *β2t* promoter-driven fluorescence would allow us to follow individual sperm, we analyzed dissected testes, individual sperm, and spermathecae of non-transgenic females inseminated by transgenic males (Figure 2). The fluorescence was sperm-specific, which indicates that the isolated 1 kb region contains all the regulatory elements responsible for sex- and tissue-specific expression, confirming another recent study in which about 700 bp of an upstream sequence were used to drive EGFP [21]. The single sperm derived from transgenic males showed the respective red or green fluorescence along the sperm tail, and the dissected spermathecae from non-transgenic females mated to transgenic males carried the transferred red or green fluorescent sperm, respectively (Figure 2). This indicates that the transgenically marked sperm does not in principle interfere with mating and paternity success, supporting the idea to use such a marker for monitoring purposes in SIT programs also in coleopteran pests and potentially use such lines for studies in reproductive biology.

To determine the sperm utilization and competition of sequentially twice-mated females, we performed a laboratory assay, in which non-transgenic *vermillion*^white^ (*v*^w^) *Tribolium* females were sequentially mated with transgenic males with first DsRed (M1_F1 and F1_M1) and second EGFP (F37_F1, F37_M2) marked sperm or vice versa. The sired progeny was followed up over twenty days after the remating (Appendix A). The results (Appendix A) confirm the notion that the females start utilizing the sperm of the last mated male most predominantly [22]. However, over time, there seems to be a balance of the use of the sperm of the first and second males, with the sperm of the first male still being used after twenty days. Since the EGFP lines (F37_F1, F37_M2) seemed to perform weaker than the DsRed lines (M1_F1 and F1_M1) in these experiments, especially when using them as second mates, we performed an outcrossing and line re-establishing procedure to overcome the potential problems of genetic bottlenecks in establishing transgenic lines.

### 2.2. Competitiveness of Sperm-Marking Lines

After outcrossing and re-establishing the homozygous sperm-marking lines, we performed competitive assays against non-transgenic males to evaluate the competitiveness of the transgenic males. For this purpose, we performed simultaneous competition in a relaxed, normal, and highly competitive line, as described in Appendix A. The results show that all the transgenic males were outperformed by the non-transgenic males (Appendix A). The reduced performance of transgenic lines has been shown previously [4] and might be due to the specific integration site of the construct or the genetic bottleneck during the establishment of the transgenic line that might remain even after some rounds of outcrossing. However, all transgenic lines sired between a fifth and almost half of the progeny, depending on the respective line (Figure 3), which indicates that the lines are reasonably competitive to be used in studies of reproductive biology. To evaluate how the different transgenic lines compete with each other, we performed simultaneous competition experiments, as outlined in Appendix A. While immediately after setting the beetles together, the EGFP sperm-marked beetles (F37_F1 and F37_M2) seemed to perform slightly better, after the removal of the males, both sperm sources were used more evenly (Appendix A). This indicates that the specific marking of the sperm probably does not cause any preference for its use, but that different lines seem to be slightly different in performing in the competitive assay.

### 2.3. Sperm Use in Consecutive Matings

With the outcrossed and re-established homozygous lines, we repeated the sperm utilization and competition in consecutive twice-mated females and followed the sired offspring over twenty days after the remating (Appendix A). The results (Figure 4) again confirm that the female’s last mate generally sires a majority of the immediate progeny [22]. However, over time, the sperm use is more balanced, with the sperm of the first male still being used after twenty days.

To determine the long-term storage of sperm within the female reproductive tract after the first and second mating, we dissected and cryosected the female reproductive tissue and evaluated the fluorescent sperm presence (Figure 5). After the first mating, the sperm seemed to occupy both the spermathecal tubules as well as the bursa copulatrix, which actually provides the fertilization set [23]. Two days after the second mating, both the first as well as the second sperm were already stored in the spermathecae. Ten days after the second mating, both types of sperm were still present and homogenously distributed (Figure 5), which is also reflected by the more equal use of both types of sperm to sire progeny at this time (Figure 4). The first-received sperm were still clearly visible in the female reproductive tract (Figure 5A^XII^) as were the secondly received sperm (Figure 5B^XII^).

## 3. Discussion

The successful establishment of differently marked lines in the red flour beetle *T. castaneum* shows that the *β2t* enhancer/promoter can also be used in Coleoptera for sperm-marking purposes. This confirms, on the one hand, the recent findings by [21] but, on the other hand, also indicates that the promoter/enhancer alone suffices and that the 5′UTR is not needed for such constructs. Droge-Young et al. (2016) [23] also used green and red fluorescently marked transgenic *T. castaneum* lines but employed the *TcProtamine-1* gene to generate complete fusion proteins for marking. In that study, however, the only red fluorescent line could not be made homozygous, hampering the progeny analysis in their reproductive biology studies to some degree. In addition, we could show that the males of at least one EGFP-sperm-marked line (F37_M2) sired almost 50% of the progeny compared with the non-transgenic males in a highly competitive situation (Figure 3), which indicates that such sperm-marked lines could be used for the monitoring of genetic pest control strategies such as SIT also in Coleoptera.

Our results regarding the reproductive biology of *T. castaneum* confirm the notion that the spermathecae are employed for the long-term storage of sperm to be used for months [24], while sperm in the bursa copulatrix is used for fertilization [23]. *T. castaneum* females can actively discard sperm after copulation [25] and also actively transfer sperm from the site of deposition, the bursa copulatrix, to the long-term storage organ, the spermathecae [26,27]. Our results confirm and clearly visualize (Figure 4 and Figure 5) that both the first as well as secondly provided sperm are stored in the spermathecal tubules and used for siring progeny for a long period of time [23]. The first occupation of the long-term storage organ by the first sperm can also explain why in multiple mating events, this first sperm is kept more predominantly than the successive sperm [28]. However, immediately after mating, mostly the newly provided sperm, which is still in the bursa copulatrix, is used for siring progeny (Figure 4), leading to the phenomenon of ‘last-male sperm precedence’ [20]. In the very promiscuous species *T. castaneum*, there are several well-documented mechanisms of cryptic female choice at pre-mating (allowing intromission), mating (spermatophore transfer and positioning), and post-mating (spermatophore ejection; remating) stages (reviewed in [15]). Our established lines could be used to further analyze the compelling process of sperm movement into the long-term storage organ, and how this is controlled by the female as part of a post-mating cryptic female choice mechanism.

## 4. Materials and Methods

### 4.1. Beetle Culture

In all the experiments included the germline transformation, unless otherwise stated, the *vermillion^white^* (*v^w^*) strain of the red flour beetle, *T. castaneum*, and was used as the non-transgenic control. Beetle stocks were fed on full grain flour supplied with 5% yeast powder in 4.5 L square plastic boxes (15 cm × 15 cm × 20 cm) with ventilation mesh fixed on the lid and were kept at 32 °C with constant light [29].

### 4.2. Isolation of the β2t Promoter Region and Cloning of the Transformation Vectors

To establish a sperm-marking system in *T. castaneum*, a 1 Kb upstream region including the 5′UTR of the *β2t* gene (TC009035; GENBANK accession number XP_969993; [16]) was amplified from the genomic DNA of the wild-type San Bernardino strain (SB-gDNA). SB-gDNA was extracted using a NucleoSpin^®^ DNA insect kit (Macherey-Nagel GmbH α Co. KG, Düren, Germany). Polymerase chain reaction (PCR) was performed using primer pair MID#4/MID#7 (Appendix A) in a 50 µL reaction consisting of water 28 µL, gDNA 2 µL, 5x Phusion buffer 10 µL, Phusion polymerase 1 µL, dNTPs 5 µL, forward primer 2.5 µL and reverse primer 2.5 µL in a program of 2 min at 98 °C, 30 sec at 98 °C, 30 sec at 68 °C, 2 min at 72 °C 35 cycles, and 10 min at 72 °C. The amplified fragment was run on a 1% agarose gel, and a distinct single band was excised and purified using a NucleoSpin^®^Gel and PCR Clean-up Kit (Macherey-Nagel GmbH α Co. KG, Düren, Germany).

The transformation vectors were constructed with a two-step cloning procedure [18]. First, the 1Kb *β2t* enhancer/promoter fragment was cloned into the shuttle vector pSL_af_tTA_af [30] by cutting both the fragment and the vector with *Nco*I*/Cla*I and ligating their ends, thereby displacing tTA-SV40 from the pSL plasmid to generate pSL_af_*β2t*_af. Then, 5 μL each of *attP* forward and reverse oligos (Appendix A) were mixed with 90µL water and placed in a heat block set at 95 °C for 5 min, after which the heat block was switched off and ramped to room temperature to anneal the oligos. pSL_af_*β2t*_af was cut using *EcoR*I*/Nco*I, and subsequently, the *attP*-annealed oligos were ligated upstream of the *β2t* P/E to generate pSL_af_*attP*_*β2t*_af. The pSL_af_*attP*_*β2t*_af was opened by *Cla*I/*Hind*III, and a 1 Kb *DsRed-SV40* or *EGFP-SV40* fragment, respectively, derived from pSL_3xP3-DsRed-SV40af [19] as *Cla*I/*Hind*III fragment and via PCR amplification from pSL_3xP3-EGFP-SV40af [19] with primer’s pair MID#103/MID#104 (Appendix A), digested by *Cla*I/*Hind*III and ligated into opened pSL_af_*attP*_*β2t*_af to obtain pSL_af*attP*-*β2t*- DsRed-SV40_af and pSL_af_*attP*-*β2t*-EGFP-SV40_af, respectively.

In the second step, the fragments *attP*_*β2t*_DsRed_SV40 and *attP*_*β2t*_EGFP_SV40 were excised out of the pSL_af_*attP*-*β2t*-DsRed-SV40_af and pSL_af_*attP*-*β2t*-EGFP-SV40_af shuttle vectors via *Asc*I digestion and inserted independently into *Asc*I-cut pBac{3xP3_EGFP_SV40af} [18] or pBac{3xP3_DsRed_SV40af} [19] transformation vectors carrying fluorescent transformation eye markers, respectively, to generate the transformation vectors pBac{3xP3_EGFP_SV40af _*attP*_*β2t*_DsRed_SV40af} (Figure 1B) or pBac{3xP3_DsRed_SV40af_*attP*-*β2t*_EGFP_SV40af} (Figure 1C), respectively.

### 4.3. Germline Transformation and Strain Establishment

Germline transformation was conducted as described [20,31]. Beetles of the *v^w^* strain were placed on white flour for 1 h, after which the embryos were sieved out and kept for an additional hour at room temperature. The embryos were washed twice with 1% Klorix solution and rinsed with clean tap water. An embryonic injection was performed using a FemtoJet^®^ Microinjector (Eppendorf, Hamburg, Germany). The needles for the microinjection were made from 10 mm×1 mm borosilicate capillaries by pulling them with a P-2000 micropipette puller (Sutter Instrument, Novato, CA, USA) with the following settings: Heat = 350, Fil = 4, Vel = 50, Del = 225, PUL = 150. The pulled needles were opened and sharpened using a Bachofer Laboratoriumsgeräte beveller (Reutlingen, Germany). For both constructs, pBac{3xP3_EGFP_SV40af _*attP*_*β2t*_DsRed_SV40af} and pBac{3xP3_DsRed_SV40af_*attP*-*β2t*_EGFP_SV40af}, 1000 (1–2 h old) embryos each were injected using a concentration of 500 ng/μL transformation vector along with 300 ng/μL helper plasmid *Tc-^m^hyPBase* [31] in an injection buffer (5 mM KCl, 0.1 mM KH_2_PO_4_, and 0.1 mM NaH_2_PO_4_ pH 6.8). The injected embryos were transferred onto an apple agar plate and sealed in a plastic box, which was then kept in an incubator at 32 °C for 2 days. After 2 days, the lid of the box containing the injected embryos was removed. Hatched larvae were individually picked with a hair-thin brush and placed on full wheat flour until pupation. Pupae (G_0_) were sexed and individually crossed to three non-injected individuals of the opposite sex. F_1_ individuals were screened for EGFP fluorescence (Figure 1B^II^) or DsRed fluorescence (Figure 1C^III^) in the eyes driven by 3xP3 promoter in both males and females using a LEICA M205 FA epifluorescence stereomicroscope with the filters EGFP-LP (excitation: ET480/40, emission: ET510 LP) or RFP (excitation: ET546/10x, emission: ET605/70m), respectively. The survival of the injected embryos and the transformation rate are documented in Appendix A for each construct.

To obtain the homozygous lines, single transgenic F_1_ individuals were outcrossed with *v^w^* beetles of the opposite sex, and the heterozygous progenies obtained were pooled for each individual F_1_, which subsequently resulted in a mixture of non-transgenic, heterozygous, and homozygous individuals. The homozygous individuals were identified based on the fluorescence intensity in the eyes and inbred. Of their progeny, 10 virgin individuals (5 males and 5 females) were randomly picked and crossed to virgin *v^w^* beetles of the opposite sex. The crosses were allowed to stand for two weeks to assure mating, after which the individuals are removed, separated, and kept in separate vials. Their offspring were allowed to grow to become adults and screened for respective fluorescence. If all the offspring found in a vial was transgenic, the founder transgenic parent was considered homozygous, kept, and crossed to establish the respective line.

To determine the position of the transgene insertion, the genomic DNA sequences flanking the 5′ and 3′ *pBac* junctions were recovered using inverse PCR. DNA was extracted from the sperm-marked individuals using a NucleoSpin^®^ DNA Insect kit (Macherey-Nagel GmbH α Co. KG, Düren, Germany). About 1 µg DNA of each transgenic line was digested with *Sau*3AI or *Msp*I for 5′ and 3’ *pBac* junctions, respectively, for 4 h at 37 °C in a 25 µL reaction. The *Sau*3AI reactions were inactivated at 65 °C for 20 min. The resulting cut DNA fragments were self-ligated in a reaction volume of 400 µL (water 333 µL, reaction 25 µL, Buffer 40 µL, and T4 DNA ligase 2 µL) at 16 °C overnight. PCRs were conducted on the self-ligated genomic DNA from several lines for both 5′ and 3′ arms using different PCR protocols. For the 5′ junction, the first PCR (3 min at 98 °C, 30 s at 98 °C, 30 s at 68 °C, 1 min at 72 °C 35 cycles, and 10 min at 72 °C) was performed using primer’s pair iPCR5′F1- 5/iPCR5′R1-6 (Appendix A). An aliquot of this first reaction served as the template for the second nested PCR (3 min at 98 °C, 30 s at 98 °C, 1 min 30 s at 72 °C, 35 cycles, and 10 min at 72 °C) carried out with primer’s pair iPCR5′F2-7/iPCR5′R2-8 (Appendix A). For the 3′ junction, just one PCR reaction was conducted using primer’s pair MFS227/MFS228 (Appendix A). The products of the PCRs were run on 1% agarose gel, and distinct bands were excised and purified using the NucleoSpin^®^Gel and PCR Clean-up Kit (Macherey-Nagel GmbH α Co. KG, Düren, Germany). The purified fragments were cloned into a pJET1.2 vector [30] and sequenced using primers pJET1.2 forward and reverse primers (Appendix A). The sequences obtained were used to search against the *T. castaneum* genome Tcas 5.2 using a BLAST search to ascertain the localization of the *piggyBac* insertion, as shown in Appendix A.

To outcross and re-establish the transgenic lines for counteracting the bottleneck effect of transgenesis, the homozygous individuals of lines M1_F1 and F1_M1 carrying p*Bac* {3xP3_EGFP-SV40af_*attP*-*β2t*_DsRed_SV40af} as well as F37_F1 and F37_M2 carrying p*Bac* {3xP3-DsRed-SV40af_attP-*β2t*_EGFP_SV40af} were singly outcrossed to the *v^w^* beetles of the opposite sex and placed on a 50 g full grain flour with 12 replications each. The setups were left to stand for a period of 10 days to mate and lay embryos, and then the adults were removed and discarded while the embryos laid were left to develop into heterozygous adults. From the 12 replications, 3 males from 6 replications each and 3 females out of the other 6 replications each were crossed to each other to set up 6 replications, which were again kept for 10 days to mate and lay eggs. The adults were subsequently taken out, and the embryos were left to develop into adults comprising non-transgenic, heterozygous, and homozygous individuals. The homozygous individuals were screened based on the high intensity of the fluorescence in the eyes. Subsequently, three homozygous males and females each were screened at the pupal stage from the 6 vials, to give 3 sets of male or female individuals, which were then crossed to give a total of 3 crosses at the end. They were left to mate and lay eggs, and after 10 days the adults were removed and kept singly. The offspring were left until they were adults and screened, and when all the sired offspring were highly fluorescent indicating homozygosity, the lines were considered homozygous again and re-established.

### 4.4. Competitive Assays

To determine sperm competition and utilization by sequentially twice-mated females, we designed an experiment, as depicted in Appendix A. Female *v^w^* beetles were first mated for 2 days with males producing either red fluorescent sperm p*Bac*{3xP3_EGFP-SV40af_*attP*-ß2t_DsRed_SV40af} or green fluorescent sperm p*Bac*{3xP3-DsRed-SV40af_*attP*-ß2t_EGFP_SV40af}. Then, the males were replaced by new males with differently marked sperm and kept for additional 2 days; then, the males were removed, and the females were placed on a new flour. The females were then moved to new flour after every 2 days for a period of 20 days. The progenies of the assay were screened for eye transformation markers to ascertain sperm competitiveness, and the numbers obtained were counted and recorded. Each setup was replicated 10 times.

To assess the direct mating competitiveness of outcrossed and re-established homozygous transgenically marked sperm lines, we competed them against non-transgenic *v^w^* males, as described in Appendix A. The completion for non-transgenic females was at a normal level of 1 × 1 × 1 (5 transgenic males × 5 non-transgenic males × 5 non-transgenic females), at a relaxed level of 1 × 1 × 2 (5 transgenic males × 5 non-transgenic males × 10 non-transgenic females), and a high level (10 transgenic males × 10 non-transgenic males × 5 non-transgenic females). All the experiments were replicated 3 times and left to stand for 4 days before the males were removed and the female transferred to new flour after every 2 days over a period of 20 days.

To assess the direct mating competitiveness between the different sperm-marked lines, we used the males from strains M1_F1 or F1_M1 (p*Bac*{3xP3_EGFP-SV40af_*attP*-ß2t_DsRed_SV40af}) and F37_F1 or F37_M2 (p*Bac*{3xP3-DsRed-SV40af_*attP*-ß2t_EGFP_SV40af}) and mated them to non-transgenic *v^w^* females. The setup was made using 5 transgenic males per strain crossed with 10 females (Appendix A). The experiment was allowed to stand initially for a period of 4 days, after which the adults were sieved, and the females were transferred to a new flour and kept for further 21 days, while the males were discarded. After 21 days, the adult females were then removed and discarded, and the offspring of both sired after 4 days or 21 days were screened via epifluorescence using the eye marker to discriminate between the strains. The experiment was replicated 3 times.

### 4.5. Microscopy

Testis, spermathecal, and single sperm imaging (Figure 2) was performed using Zeiss Imager Z2 equipped with 2 cameras: Axiocam 506 mono for EGFP-LP (excitation: ET480/40, emission: ET510 LP) or DsRed (excitation: 533–558, emission: 570–640) and Axiocam 305 color for brightfield. The dissection of the testis or spermatheca was carried out by placing the adults on ice for 10 min. The male or female individual to be dissected was later picked up and placed in a pool of insect saline solution, and by slightly squeezing the thorax of the individual with a pair of forceps, the genitalia was exposed; using another pair of forceps, the exposed genitalia was pulled out and moved into an ice-cold 1xPBS solution before fixation. The testes or spermathecae were fixed as described [32] with some modification, whereby the tissues were made to stand for 30 min using DAPI (4′,6-Diamidino-2-phenylindole, dihydrochloride) for nuclei staining. Afterward, they were mounted in 70% glycerol for image capturing. To obtain the single sperms, the dissected testes were put in 1.5µL plastic tubes and ground several times using a pestle, and vortexed in between the grinding process to facilitate single sperms going into the solution. Afterward, the dissected spermathecae and/or testes were placed on a slide singly and further separated carefully before mounting them for microscopy. The suspended sperms in a solution were pipetted on a slide and covered with a coverslip.

The spermathecae of single or twice-mated females with EGFP and/or DsRed transgenically sperm-marked male individuals (Figure 5) were dissected and embedded in an OCT embedding matrix (Carl Roth GmbH + Co. KG, Karlsruhe, Germany) according to manufacturer’s instructions and kept in a −80 °C freezer. The spermathecae were cryosected in a cryostat (Leica CM1950, Leica Biosystems Nussloch GmbH, Nussloch, Germany) machine set at 8 mm.

## Figures and Tables

**Figure 1 ijms-23-12594-f001:**
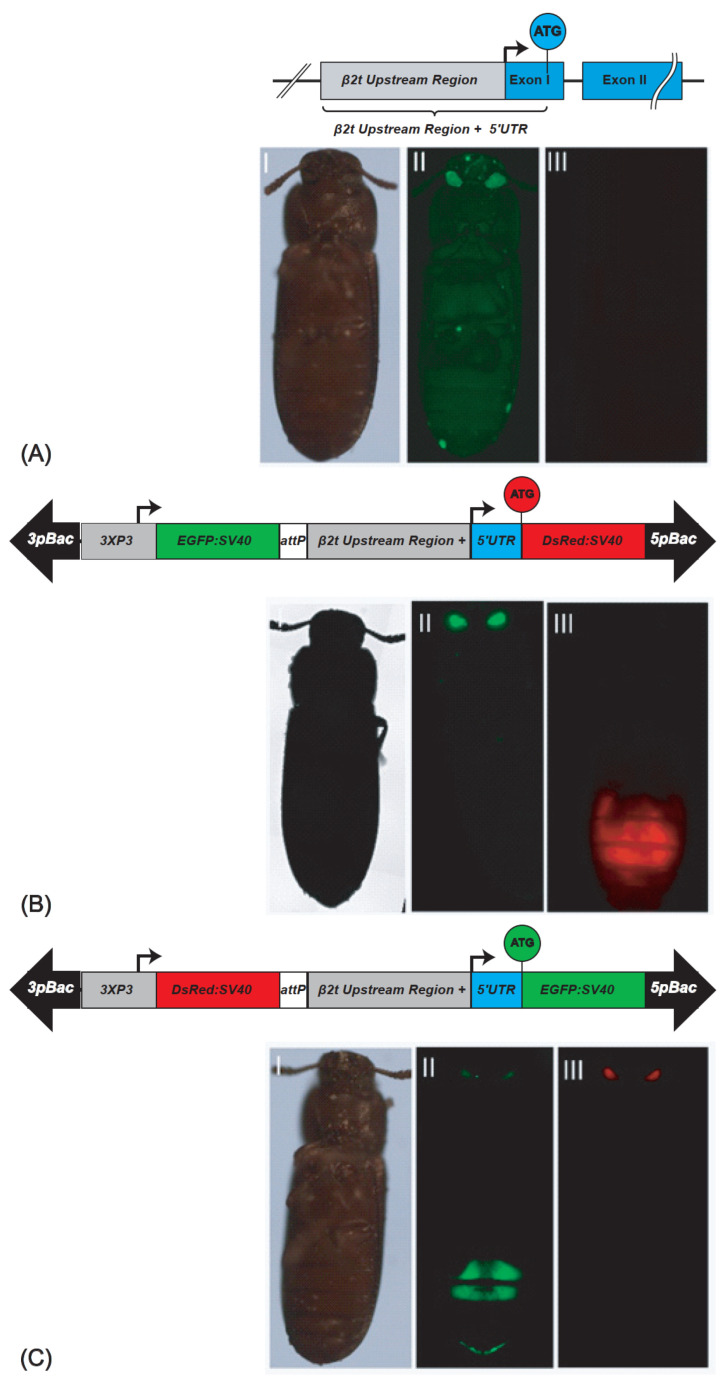
*β2t* gene and derived sperm-marking constructs: (**A**) schematic depiction of the endogenous *β2t* gene (TC009035) showing the upstream region in grey, the exons in blue, and the intron as black line. (A^I^–A^III^) non-transgenic male photographed under white light, EGFP filter, and DsRed filter; (**B**) pBac{3xP3_EGFP_SV40af _*attP*_*β2t*_DsRed_SV40af}: construct for red fluorescent sperm. (B^I^–B^III^) *β2t*-DsRed-SV40 carrying male photographed under white light, EGFP filter, and DsRed filter; (**C**) pBac{3xP3_DsRed_SV40af_*attP*-*β2t*_EGFP_SV40af}: construct for green fluorescent sperm. (C^I^–C^III^) *β2t*-EGFP-SV40 carrying male photographed under white light, EGFP filter, and DsRed filter. The opposite fluorescence in the eyes is due to the respective transformation markers 3xP3-EGFP or 3xP3-DsRed.

**Figure 2 ijms-23-12594-f002:**
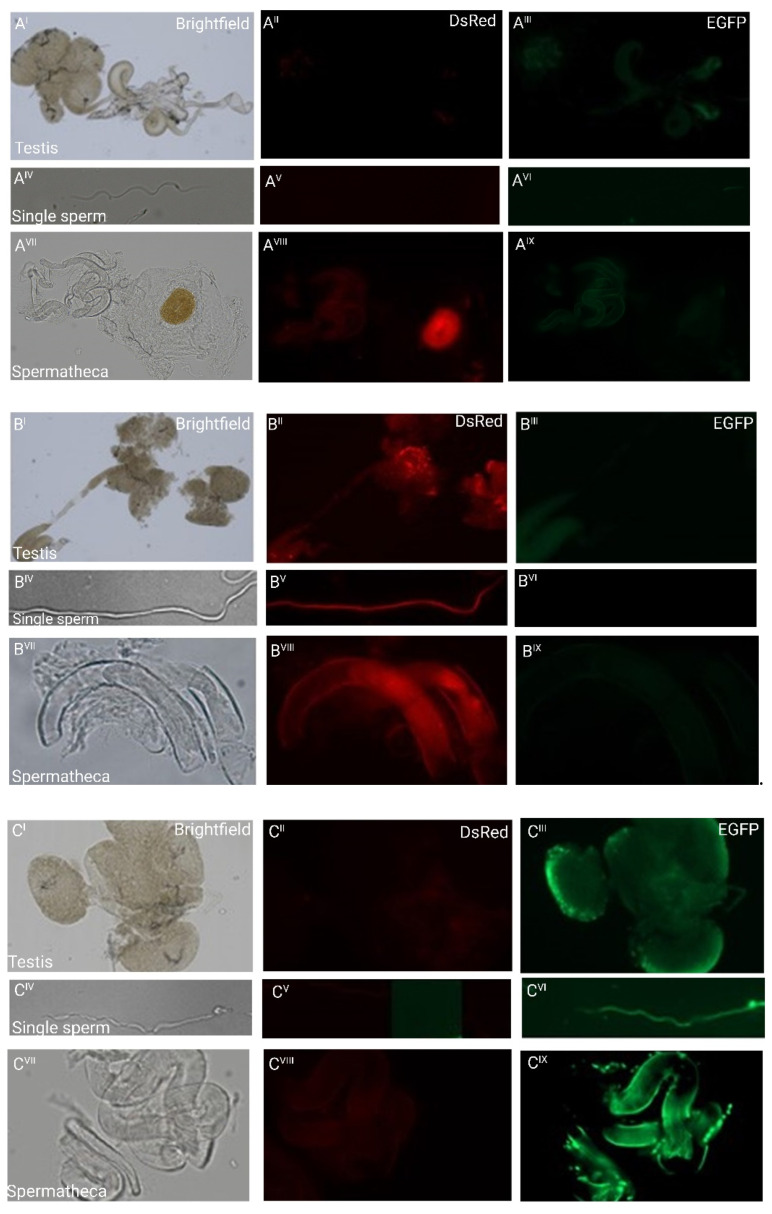
Fluorescent sperm detection in dissected testes (I–III), as individual sperm (IV–VI), and spermathecae (VII–IX) of non-transgenic females inseminated by transgenic males: (**A**) weak green autofluorescence in the testes of non-transgenic males and red autofluorescence of the chitinous O-ring structure to the right of the spermathecal tubules of a non-transgenic female inseminated by a non-transgenic male. No other fluorescence patterns are observed; (**B**) the testes and single sperm from a transgenic male carrying pBac{3xP3_EGFP_SV40af _*attP*_*β2t*_DsRed_SV40af} and the spermathecal tubules of a non-transgenic female mated to such a male show strong red fluorescence only; (**C**) the testes and single sperm from a transgenic male carrying pBac{3xP3_DsRed_SV40af _attP_*β2t*_EGFP_SV40af} and the spermathecal tubules of a non-transgenic female mated to such a male show strong green fluorescence only.

**Figure 3 ijms-23-12594-f003:**
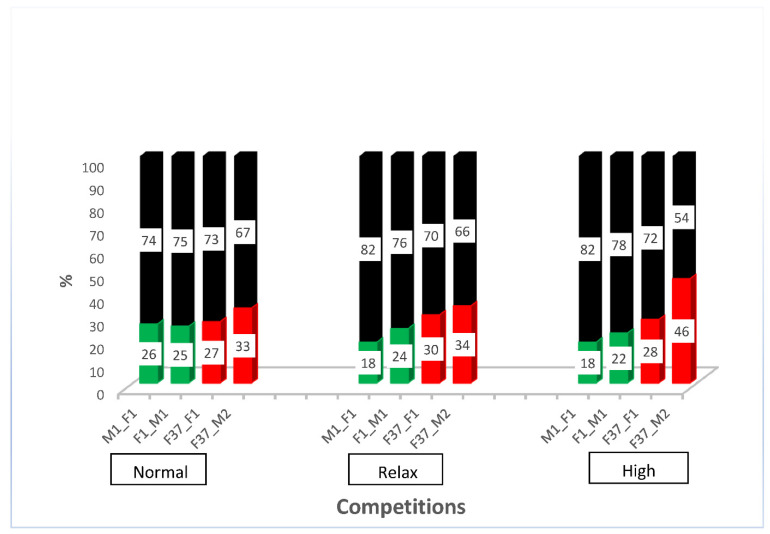
Simultaneous competition of transgenic males compared with non-transgenic males. Percentage of offspring sired by non-transgenic (black) and transgenic males (green or red, respectively) in a relaxed (5 transgenic males × 5 non-transgenic males × 10 non-transgenic females), normal (5 transgenic males × 5 non-transgenic males × 5 non-transgenic females), and high competition (10 transgenic males × 10 non-transgenic males × 5 non-transgenic females).

**Figure 4 ijms-23-12594-f004:**
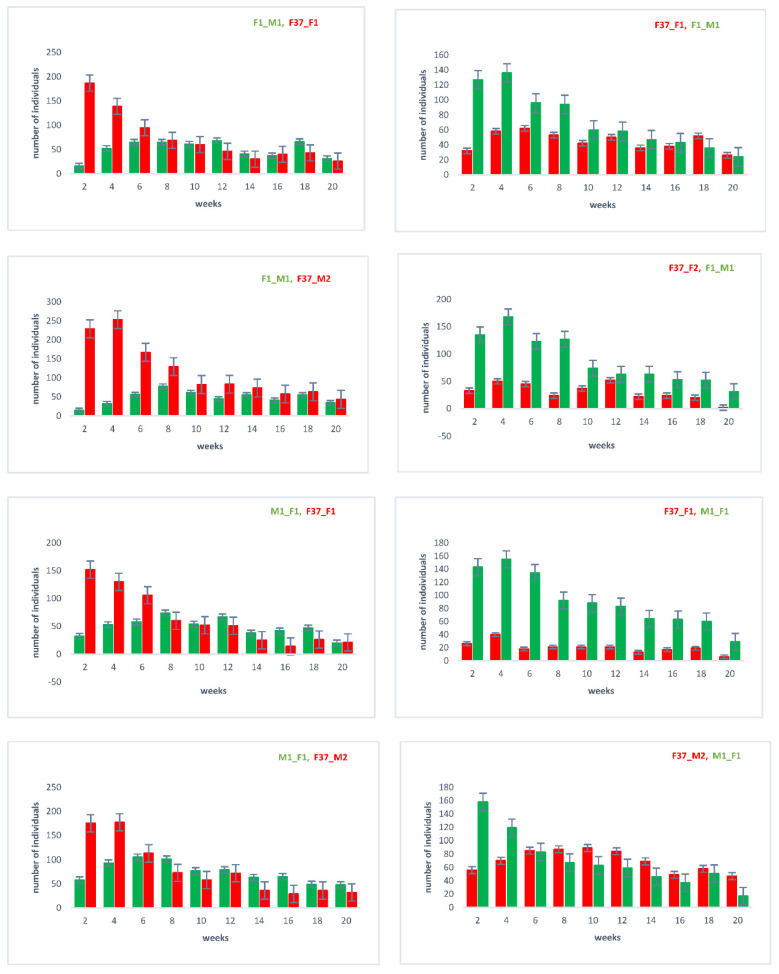
Sperm utilization and competition of sequentially twice-mated females. Mating events were performed as indicated in Appendix A, with the order of the respective lines indicated above on each chart. In the progeny, the number of individuals showing the respective eye marker (EGFP or DsRed; see Figure 1) was counted, which indirectly indicates the used sperm.

**Figure 5 ijms-23-12594-f005:**
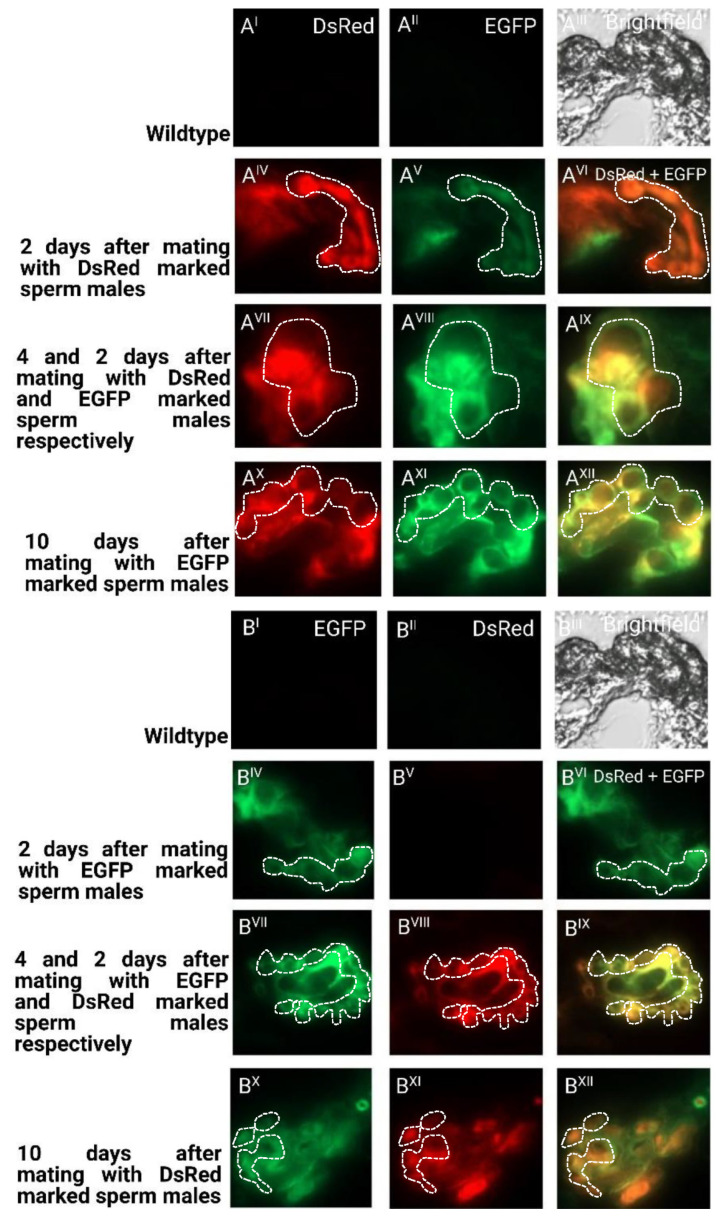
Cryosections of female reproductive tissue with fluorescent sperm after successive mating events: (I–III) non-transgenic females mated to non-transgenic males as negative control regarding autofluorescence; (IV–VI) two days after mating with first male before mating with second male; (VII–IX) two days after second mating; (X–XII) ten days after second mating. Pictures were taken for detection of EGFP (I, IV, VII, X), for detection of DsRed (II, V, VIII, XI), and overlay of channels (VI, IX, XII); (**A**) mating first with males providing DsRed marked sperm and consecutively with males providing EGFP marked sperm; (**B**) mating first with males providing EGFP-marked sperm and consecutively with males providing DsRed-marked sperm. The tubular structures of the spermathecae are indicated with a dashed line.

## Data Availability

Not applicable.

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
