# Peer review of "Competitive Sperm-Marked Beetles for Monitoring Approaches in Genetic Biocontrol and Studies in Reproductive Biology"

_ijms, 2022, doi:10.3390/ijms232012594_

Round 1
Reviewer 1 Report
This is an excellent study that describes the generation of transgenic beetles that can be used for sperm monitoring in mating competition experiments.
Minor comments:
1) Can it be clarified that the β2-tubulin promoter is testis/sperm-specific?
2) How was the promoter region selected since no transcription start site was defined?
3) The graphs presented in Figure 4 lack labeling of X- and Y-axes. The authors should also include a presentation of the experimental set-up as in Figure S1.
4) Figure 5 (fluorescent images): bursa copulatrix and spermatheca should be indicated (and explained how these were identified).
5) The authors should read the text again carefully and correct spelling mistakes and missing words.
Author Response
1) Can it be clarified that the β2-tubulin promoter is testis/sperm-specific?
We have added a sentence to the restriction to the testes, when looking at adult beetles and clarified another sentence to better state the sperm-specificity in dissected tissue and in addition referenced again Khan et al., 2021, who also observed the same specificity.
2) How was the promoter region selected since no transcription start site was defined?
Based on transcriptomics data, the 5#UTR is 121 base pairs as written in the manuscript. For making the construct we chose a 1 kb fragment that included the 5’UTR and upstream sequences. We have clarified this now in the writing and also redid Figure 1 to give an exact representation of the constructs in scale.
3) The graphs presented in Figure 4 lack labeling of X- and Y-axes. The authors should also include a presentation of the experimental set-up as in Figure S1.
The values of the X- and Y-axes have been added (similarly also in the Supplementary Figures, were they were unlabeled). The experimental set-up was and is presented in Supplementary Figure S1 as indicated in the Figure legend!
4) Figure 5 (fluorescent images): bursa copulatrix and spermatheca should be indicated (and explained how these were identified).
The spermatheca were identified by their tubular composition and are now indicated by a dashed line. The focus of the manuscript is on the long term storage in the spermathecae. Therefore more focus was put on this structure, which is now also better indicated in the text.
5) The authors should read the text again carefully and correct spelling mistakes and missing words.
We have reread the manuscript carefully and corrected the spelling mistakes and missing words, we detected. Thanks for pointing this out to us.
Reviewer 2 Report
The manuscript titled “Competitive sperm-marked beetles for monitoring approaches in genetic biocontrol and studies in reproductive biology” provide sperm-marked lines with fluorescence for the red flour beetle. There is a historical bias of using Sterile Insect Technique (SIT) towards mostly Diptera and Lepidoptera, this study aimed to control Coleoptera pests through SIT. However, transgenic lines marked with EGFP or DsRed fluorescence were employed for sperm movement analysis and sperm competitiveness determination. It’s not clear whether this technology could be applied for pest control, I suggest the authors emphasize less on SIT in introduction, and pay more attention to the importance of this work. Below are my comments.
1. SIT was mentioned many times in introduction, but there is no information about transgenic SIT lines generated in this work. It would be great to make it more clear whether there is a transgenic SIT line in this work. If there is a transgenic line, the methods for generating SIT lines should be mentioned clearly. If not, it would be better to reintroduce the importance of this work.
2. In figure 1, it would be great to exhibit more information in vector constructs. It’s not clear that these sperm-marked lines were generated through homologous recombination or casual insertion. If homologous recombination was carried out, the reason for sperm competitiveness difference between EGFP sperm marked beetles and DsRed sperm marked beetles should be discussed (line 164-168). If transgenic lines were generated through casual insertion, more lines are needed.
3. More information is needed for legends. Such as figure 4, there is no explanation for the number of x-axis.
4. After outcrossing transgenic lines with wild type lines, all transgenic line sire between a fifth and almost half of the progeny theoretically, it is also mentioned this way (line 163-164). In figure 3, the ratio of transgenic lines in progeny is less than 50%, this should be included in discussion why these transgenic lines are less competitive.

Author Response
- SIT was mentioned many times in introduction, but there is no information about transgenic SIT lines generated in this work. It would be great to make it clearer whether there is a transgenic SIT line in this work. If there is a transgenic line, the methods for generating SIT lines should be mentioned clearly. If not, it would be better to reintroduce the importance of this work.
We have re-arranged the introduction and abstract to start no with reproductive biology studies and only then pint to SIT, which we think is an important point for doing this type of research. Therefore, we kept Sit but not as the primary focus. We have also now stated clearly that Tribolium is not a species for SIT and thus developing applicable lines in this species is not intended. However, Tribolium castanuem is a modle organism for coleopterans and thus our study provides a proof-of-principle, that such line can be generated in Coleoptera.
2. In figure 1, it would be great to exhibit more information in vector constructs. It’s not clear that these sperm-marked lines were generated through homologous recombination or casual insertion. If homologous recombination was carried out, the reason for sperm competitiveness difference between EGFP sperm marked beetles and DsRed sperm marked beetles should be discussed (line 164-168). If transgenic lines were generated through casual insertion, more lines are needed.
We have now changed Figure 1 to clearly indicate the constructs used. As already previously mentioned in the text, we used-piggyBac-mediated germline transformation, which however is now also more obviously depicted in the Figure.
We don’t think there is a clear identifiable difference between DsRed- or EGFP-marked lines. Only Supplementary Figure S6 points to some initial outperforming of the EGFP sperm-marked beetles. However, we phrased that very softly and added now even a sentence to make sure that we think that this is probably rather line specific than construct specific. Altogether, we generated 18+4=22 transgenic lines based on F1-progeny (now indicated in added Supplementary Table S2). For the reproductive biology analyses, we used lines that could be homozygoused, which limited our selection.
More information is needed for legends. Such as figure 4, there is no explanation for the number of x-axis.
The values of the X- and Y-axes have been added (similarly also in the Supplementary Figures, were they were unlabeled). More information was also added to the Figure legend.
4. After outcrossing transgenic lines with wild type lines, all transgenic line sire between a fifth and almost half of the progeny theoretically, it is also mentioned this way (line 163-164). In figure 3, the ratio of transgenic lines in progeny is less than 50%, this should be included in discussion why these transgenic lines are less competitive.
We have added a sentence discussing the possibilities of why transgenic lines are often less competitive as had been shown before.
Round 2
Reviewer 2 Report
No more questions or suggeations